# Impact of mRNA and Inactivated COVID-19 Vaccines on Ovarian Reserve

**DOI:** 10.3390/vaccines13040345

**Published:** 2025-03-24

**Authors:** Enes Karaman, Adem Yavuz, Erol Karakas, Esra Balcioglu, Busra Karaca, Hande Nur Doganay, Koray Gorkem Sacinti, Orhan Yildiz

**Affiliations:** 1Department of Obstetrics and Gynecology, Faculty of Medicine, Nigde Omer Halisdemir University, 51240 Nigde, Turkey; 2School of Health Sciences, Cappadocia University, 50400 Nevsehir, Turkey; 3Op. Dr. Erol Karakas Gynecology-Obstetrics-Genital Aesthetics and Sexual Health Clinic, 38140 Kayseri, Turkey; 4Department of Histology and Embryology, Faculty of Medicine, Erciyes University, 38030 Kayseri, Turkey; 5Hakan Cetinsaya Good Clinical Practice and Research Center, 38030 Kayseri, Turkey; 6Obsterics annd Gynecology Clinic, Nigde Omer Halisdemir University Training and Research Hospital, 51100 Nigde, Turkey; 7Obstetrics and Gynecology Clinic, Aksaray Training and Research Hospital, 68200 Aksaray, Turkey; 8Deparment of Clinical Microbiology and Infectious Disease, Faculty of Medicine, Erciyes University, 38030 Kayseri, Turkey

**Keywords:** rat, mRNA vaccine, inactivated COVID-19 vaccines, ovarian reserve, follicular atresia

## Abstract

**Objectives**: This study aimed to elucidate the effects of messenger RNA (mRNA) and inactivated coronavirus disease 2019 (COVID-19) vaccines on ovarian histology and reserve in rats. **Methods**: Thirty female Wistar albino rats, aged 16–24 weeks, were randomly divided into three groups (*n* = 10): control, mRNA vaccine, and inactivated vaccine groups. Each vaccine group received two doses (on day 0 and day 28) at human-equivalent doses. Four weeks post-second vaccination, ovarian tissues were harvested for analysis. **Results:** Immunohistochemical analysis was performed to evaluate the expression of transforming growth factor beta-1 (TGF-β1), vascular endothelial growth factor (VEGF), caspase-3, and anti-Müllerian hormone (AMH) in ovarian follicles. Both vaccines induced significant increases in TGF-β1, VEGF, and caspase-3 expression, with more pronounced effects in the mRNA vaccine group. Conversely, AMH expression in the granulosa cells of primary, secondary, and antral follicles showed marked reductions (*p* < 0.001). The counts of primordial, primary, and secondary follicles decreased significantly in the inactivated vaccine group relative to controls and further in the mRNA vaccine group compared to the inactivated group (*p* < 0.001). Additionally, the mRNA vaccine group exhibited a decrease in antral and preovulatory follicles and an increase in atretic follicles compared to the other groups (*p* < 0.05). The serum AMH level was diminished with the mRNA vaccination in comparison with the control and inactivated groups. **Conclusions**: Our findings suggest that both mRNA and inactivated COVID-19 vaccines may detrimentally impact ovarian reserve in rats, primarily through accelerated follicular loss and alterations in apoptotic pathways during folliculogenesis. Given these observations in a rat model, further investigations into the vaccines’ effects on human ovarian reserve are needed.

## 1. Introduction

COVID-19, attributed to the novel Severe Acute Respiratory Syndrome Coronavirus-2 (SARS-CoV-2), first identified in December 2019, has precipitated a global health crisis [1]. Following the disclosure of the viral genome by Chinese scientists in January 2020, originating from initial cases of pneumonia in Wuhan, the global scientific community rapidly mobilized to develop countermeasures [2]. In December 2022, twelve vaccines had received World Health Organization (WHO) approval, employing diverse technological approaches [3]. COVID-19 vaccines are still an effective approach to protect against COVID-19 and are recommended for women of reproductive age [4]. Despite their proven efficacy, emerging reports have noted menstrual irregularities following administration of COVID-19 vaccines utilizing mRNA and adenovirus vectors, although some studies report no significant association [5,6,7]. This has prompted investigations into potential links between these vaccines and reproductive health disturbances [8,9,10,11].

Successful reproduction hinges on a complex interplay of biological processes, with the ovaries and ovarian follicles playing critical roles [12]. The primary functions of the ovaries include the production and periodic release of oocytes, as well as the secretion of steroid hormones [13,14]. Ovarian follicles, the functional unit of the ovaries, consist of oocytes, granulosa cells (GCs), and theca cells (TCs) [14]. Follicular development, which underpins the physiological processes of estrus and ovulation [15], initiates with the transformation of primordial follicles into the growing follicle pool, advancing through progressively mature stages [16]. This growth either culminates in the emergence of a dominant follicle, which releases a mature oocyte, or leads to follicular atresia [17]. The transition from primordial to primary follicles, known as initial recruitment, is governed by a suite of paracrine factors independently of follicle-stimulating hormone (FSH). During cyclic recruitment, elevated levels of circulating FSH promote the survival of a cohort of antral follicles (2–5 mm in diameter), preventing their apoptotic demise [18].

The functionality of the ovaries and ovarian follicles is sustained through continuous angiogenesis, the process by which new blood vessels develop from pre-existing vasculature [19]. VEGF is posited as a critical mediator in the vascular recruitment necessary for follicle nourishment [20]. Additionally, signaling via the transforming growth factor-beta (TGF-β) family, which includes AMH and TGF-β isoforms (TGF-β1-3), plays a pivotal role in regulating ovarian folliculogenesis, influenced by both intra- and extraovarian factors [21]. Anti-Müllerian hormone (AMH), secreted by the granulosa cells (GCs) of developing follicles, is known to suppress the activation of primordial follicles in mice [22] and to moderate FSH-driven follicular growth [23]. TGF-β1, in particular, markedly inhibits the activation of primordial follicles and induces oocyte apoptosis [24]. Apoptosis, the primary cellular process leading to ovarian follicular atresia, prominently features caspase-3 as its major effector enzyme [25].

Previous observational studies analyzing blood samples from women of reproductive age with COVID-19 have reported inconsistent results regarding sex hormones and ovarian reserve [26]. On the other hand, the majority of data regarding the impact of COVID-19 vaccines on female fertility derive from investigations into assisted reproductive outcomes among infertile women [27]. However, the direct effects of COVID-19 vaccines on ovarian follicles and ovarian reserve have not yet been thoroughly elucidated within ovarian tissue itself. In our study, we aim to investigate the effects of mRNA and inactivated COVID-19 vaccines on ovarian follicles and ovarian reserve by quantifying the expression of TGF-β1, VEGF, caspase-3, and AMH, as well as the stages of follicular development in rat ovarian tissue post-vaccination.

## 2. Materials and Methods

### 2.1. Ethical Declaration

The study protocol received ethical approval from the Erciyes University Animal Experiments Local Ethics Committee (Approval No. 22/041, Date: 2 March 2022) and was conducted at the Erciyes University Experimental Research Application and Research Center (DEKAM).

### 2.2. Experimental Animals and Care

Thirty female Wistar albino rats, aged 16–24 weeks and weighing 240 ± 30 g, were used in this study. All experimental procedures were conducted in accordance with the Universal Declaration of Animal Rights. Rats were housed in climate-controlled rooms set to a temperature of 22 ± 1 °C with a 12 h light/dark cycle. A maximum of four rats were kept per cage. They had ad libitum access to standard pellet feed and tap water throughout the study.

### 2.3. Vaccination Procedure

The rats were randomly divided into three groups of ten animals each: Group I (control group), Group II (mRNA vaccine group) and Group III (inactivated vaccine group). In the control group rats, only 0.9% NaCl solution, the diluent of the vaccines, was administered intramuscularly at a dose of 0.4 mL on the vacc ine administration days (days 0 and 28). For the vaccinated groups, the dosing and schedule mirrored those used in humans. In the mRNA vaccine group, the BNT162b2 mRNA vaccine (Pfizer-BioNTech/Comirnaty, Mainz, Germany) was reconstituted with 1.8 mL of 0.9% NaCl solution and administered intramuscularly in two doses of 0.3 mL each, on day 0 and day 28. Similarly, the inactivated vaccine group received two intramuscular doses of the CoronaVac vaccine (Sinovac Life Sciences, Beijing, China), each dose being 0.5 mL, on the same schedule of day 0 and day 28. The vaccine dose used in our study was determined in accordance with established regulatory guidelines, including the 2006 Food and Drug Administration (FDA) Guidance (Guidance for Industry—Considerations for Developmental Toxicity Studies for Preventive and Therapeutic Vaccines for Infectious Disease Indications) [28], the 2014 World Health Organization (WHO) guidelines (Guidelines on the Non-Clinical Evaluation of Vaccine Adjuvants and Adjuvanted Vaccines, WHO Technical Report Series, TRS 987, Annex 2) [29], and the 2020 European Medicines Agency (EMA) ICH Guideline S5 (R3) (Detection of Toxicity to Reproduction for Human Pharmaceuticals) [30]. These guidelines recommend, where feasible, that animals receive the maximum human dose regardless of body weight.

### 2.4. Tissue Collection

Four weeks following the administration of the second vaccine dose, all rats were euthanized via cervical dislocation under general anesthesia, which was induced using 50 mg/kg of 10% ketamine hydrochloride (Ketasol; Richter Pharma, Wels, Austria) combined with 10 mg/kg of 2% Xylazine (Rompun; Bayer Healthcare, Berlin, Germany). Subsequent to euthanasia, bilateral oophorectomy was performed on each rat.

### 2.5. Histological Procedure

Ovarian tissues were first fixed in a 10% formaldehyde solution, then dehydrated through a graded series of alcohols, cleared in xylene, and finally embedded in paraffin. Sections of 5 µm thickness were cut using a microtome. Histological sections underwent routine Hematoxylin and Eosin (H&E) and Masson’s Trichrome (MT) staining. All slides were examined under an Olympus BX51 light microscope (Tokyo, Japan) and images were captured using an Olympus DP72 digital camera. Histological analysis was conducted by an experienced histologist who was blinded to the study group allocations [31].

### 2.6. Histological Analysis and Follicle Classification

Serial sections were prepared from the ovarian tissues embedded in paraffin blocks. One in every ten sections was stained with Hematoxylin and Eosin (H&E) for follicle counting. Only follicles with clearly visible nuclei were counted to ensure accuracy. The evaluation also considered the integrity of the basement membrane, the theca layer, the arrangement of granulosa cells (GCs), and the oocyte structure. Based on established criteria, follicles were categorized into six types: primordial (PrmF), primary (PriF), secondary (SF), antral (AF), preovulatory (PF) [32], and atretic follicles (ATF) [13].

Follicles in which the oocyte is surrounded by a single row of squamous (flattened) GCs were classified as PrmF; follicles in which the oocyte is surrounded by a single row of GCs composed predominantly of cuboidal GCs with some squamous GCs or only cuboidal GCs were classified as PriF; without a visible antrum, follicles surrounded by more than one layer of cuboidal GCs were counted as SF; early AFs with one or two small follicular fluid areas (antrum) and AFs with a large antrum together were counted as AF; and the largest of the follicular types and follicles with a defined layer of cumulus granulosa cells were counted as PF [32]. In addition, follicles with irregularly degenerated oocytes and pyknotic GCs were also considered as ATF [13].

### 2.7. Immunohistochemical Analysis

Immunohistochemical staining was performed using the avidin–biotin peroxidase method, as outlined in the ImmunoCruz™ Staining System protocol (Santa Cruz Biotechnology Inc., Santa Cruz, TX, USA). Sections of 5 microns were deparaffinized, rehydrated, and initially incubated in phosphate-buffered saline (PBS) thrice for 5 min each at room temperature. To suppress endogenous peroxidase activity, slides were treated with 3% hydrogen peroxide in methanol for 30 min. Non-specific binding was blocked by incubating the sections in 10% normal goat serum for 10 min at room temperature. The sections were then incubated overnight at 4 °C with primary antibodies against AMH (Santa Cruz Biotechnology), VEGF (Santa Cruz Biotechnology Inc.), TGF-β1 (Elabscience, Houston, TX, USA), and caspase-3 (Cell Signaling Technology, Danvers, MA, USA). This was followed by a 15 min incubation with biotinylated secondary antibody at room temperature and subsequent treatment with Streptavidin Peroxidase conjugate for another 15 min. Visualization was achieved with DAB (3,3′-diaminobenzidine tetrahydrochloride) substrate for 15 min, followed by counterstaining with Mayer’s hematoxylin. Sections were dehydrated through an alcohol series, cleared in xylene, and mounted with Entellan (Merck, Darmstadt, Germany). Imaging was performed using an Olympus BX 51 light microscope.

Quantitative analysis was conducted on images of ten different microscopic fields at 40× magnification for each experimental group, using the ImageJ.JS v0.5.8 Software program (National Institutes of Health, Bethesda, MD, USA). This analysis focused on the immunoreactivity intensities of TGF-β1, VEGF, and caspase-3 in ovarian follicles, and AMH in primary (PriF), secondary (SF), and antral follicles (AFs). Additionally, apoptosis was assessed by counting caspase-3-positive cells in ten randomly selected fields [31].

### 2.8. Measurement of Serum AMH Levels

The enzyme-linked immunosorbent (ELISA) kit (201-11-1246, Baoshan District, Shanghai, China) was used to measure the serum anti-Müllerian hormone (AMH) levels. The analysis was performed according to the manufacturer’s instructions. The quantities were measured at 450 nm in a micro ELISA reader (BioTek ELx800, Bio-Tek Instruments Inc., Winooski, VT, USA) [33].

### 2.9. Statistical Analysis

All statistical evaluations were conducted using the SPSS software (Version 24). The normality of data distribution within the groups was assessed using the Kolmogorov–Smirnov and Shapiro–Wilk tests. For variables that conformed to a normal distribution, differences between groups were analyzed using One-way Analysis of Variance (ANOVA), with post-hoc comparisons conducted via the Tukey test where significant differences were found. For variables not adhering to a normal distribution, intergroup differences were examined using the Kruskal–Wallis test, followed by the Mann–Whitney U test for post-hoc analysis when required. A *p*-value of less than 0.05 was considered statistically significant [33].

## 3. Results

When the H&E and MT staining results were evaluated under light microscopy, the ovarian tissue of the control group showed a normal appearance, with a cortex containing follicles at different stages of development on the outside and a medulla layer with loose connective tissue and an extensive vascular network on the inside (Figure 1A,D). In the mRNA vaccine group, atrophic areas in the ovarian medulla and an increase in the number and enlargement in the diameter of blood vessels were observed (Figure 1B,E). Similar findings were observed in the ovarian medulla in the inactivated vaccine group, although not as much as in the mRNA vaccine group (Figure 1C,F).

The mean counts of primordial, primary, and secondary follicles were significantly reduced in the inactivated vaccine group compared to controls, and in the mRNA vaccine group compared to the inactivated vaccine group (*p* < 0.001). Similarly, mean counts of antral follicles (AFs) and preovulatory follicles (PFs) were lower in the mRNA vaccine group than in controls (*p* < 0.001). Conversely, the mean number of atretic follicles (ATFs) increased in the mRNA vaccine group compared to both the control and inactivated vaccine groups (*p* < 0.05) (Table 1).

### 3.1. TGF-β1, VEGF, Caspase-3, and AMH Expression

In the control group, TGF-β1 expression was weakly positive, with few positive cells observed in GCs, TCs, and corpus luteum (CL) cells. In contrast, the mRNA vaccine group exhibited a markedly higher TGF-β1 expression across all regions compared to both the control and inactivated vaccine groups (Figure 2).

VEGF expression was lowest in the control group and highest in the mRNA vaccine group. Minimal VEGF expression was noted in the germinal epithelium, vascular wall, oocyte, GCs, and CL cells of the control group. In the mRNA vaccine group, VEGF expression was significantly elevated, particularly in the germinal epithelium, oocytes, GCs, the theca layer, vascular wall, CL, and intrafollicular fluid (Figure 3).

Caspase-3 expression was increased in both vaccine groups compared to controls, with a more pronounced elevation in the mRNA vaccine group, characterized by intense staining in oocytes, GCs, and interstitial stromal cells. Caspase-3-positive cell counts were significantly higher in the mRNA and inactivated vaccine groups than in controls (Figure 4).

AMH expression in all groups decreased progressively from primordial to preovulatory follicles (Figure 5). AMH expression in GCs of primordial and secondary follicles was lower in the inactivated vaccine group compared to controls and further reduced in the mRNA vaccine group (*p* < 0.001). In antral follicles, AMH expression was lower in both vaccine groups compared to controls (*p* < 0.001), with the mRNA group showing a greater, though statistically insignificant, decrease compared to the inactivated vaccine group (*p >* 0.05).

Conversely, the immunostaining intensity of VEGF, TGF-β1, and caspase-3 was significantly higher in the inactivated vaccine group than in controls and further elevated in the mRNA vaccine group (*p* < 0.001). The comparative immunostaining intensities of AMH, VEGF, TGF-β1, and caspase-3 across groups are summarized in Table 2.

### 3.2. Serum AMH Levels

After the experiment, serum AMH levels were analyzed, and it was found that the mRNA vaccine significantly reduced the AMH level compared to the control and inactivated vaccine groups (*p* < 0.001) (Figure 6). In the group administered the inactive vaccine, AMH levels were close to the control group; therefore, there was no difference between these two groups.

## 4. Discussion

In our study, we investigated the effects of mRNA and inactivated COVID-19 vaccines on ovarian follicles and ovarian reserve in a rat model, following the dose recommendations outlined by the Food and Drug Administration (FDA) [28], the World Health Organization (WHO) [29], and the European Medicines Agency (EMA) [30] for assessing the toxic effects of vaccines in animal studies. Our findings revealed that follicular VEGF, TGF-β1, and caspase-3 expression, as well as the mean atretic follicle (ATF) count, were elevated in the vaccine groups compared to the control group, whereas serum anti-Müllerian hormone (AMH) levels decreased across all follicular forms, except for the mean ATF count. These results consistently indicate that both COVID-19 vaccines, particularly the mRNA vaccine, may exert toxic effects on ovarian follicles and ovarian reserve in rats.

The members of the TGF-β superfamily are critical developmental morphogens [21] and key regulators of folliculogenesis, oocyte meiosis, and stromal remodeling [34]. Connective tissue growth factor (CTGF), a paracrine regulator involved in mitosis, angiogenesis, cell migration, and extracellular matrix remodeling, is predominantly expressed in granulosa cells, with TGF-β1 shown to upregulate CTGF expression in rat granulosa cells [35]. Elevated serum TGF-β1 levels have also been reported in COVID-19 infections, further highlighting its involvement in systemic and local tissue response [36]. In a radiotherapy-induced premature ovarian failure model, TGF-β1 expression was predominantly observed in granulosa cells of atretic follicles and the corpus luteum, with negligible expression in theca cells. The optical density of TGF-β1-positive cells was markedly increased in irradiated ovaries, indicating a strong localized response to tissue damage [37]. In our study, TGF-β1 expression was weakly positive in the control group, with few positive cells identified in granulosa cells, theca cells, and the corpus luteum. In contrast, the mRNA and inactivated vaccines caused markedly elevated TGF-β1 expression in the ovarian tissues. These findings suggest that COVID-19 vaccines, particularly mRNA vaccines, may enhance TGF-β1-mediated processes such as mitosis, angiogenesis, cell migration, and extracellular matrix remodeling in ovarian tissue, particularly within follicles composed of diverse cell types and structures. It is seen that vaccination triggers systemic inflammatory responses, leading to increased cytokine production, including IL-6, TNF-α, IL-1 β [38] and TGF-β1 [36], which play key roles in folliculogenesis, angiogenesis, and atresia [39]. Our findings indicate elevated TGF-β1 expression, which has been associated with ovarian tissue remodeling and fibrosis [21].

VEGF plays an essential role in regulating both physiological and pathological ovarian angiogenesis. VEGF can foster a proangiogenic environment that supports follicular development or shift to an antiangiogenic state during atresia, thereby inhibiting follicular progression [40]. VEGF has been demonstrated to stimulate cell proliferation in GCs and AFs, inhibit apoptosis, and suppress caspase-3 activation in AFs [41]. In untreated adult rats, VEGF-A staining was reported to be absent or weak in GCs and TCs of preantral follicles and early AFs but stronger in GCs and TCs of PFs [42]. In our study, VEGF expression was minimal in oocytes and GCs in the control group but notably stronger in oocytes, GCs surrounding the oocyte, intrafollicular fluid, and theca layer in the mRNA vaccine group. Based on earlier findings [42], which observed the highest VEGF expression in PFs, we expected VEGF expression to be highest in the control group, which had the largest number of PFs. However, VEGF expression was lowest in the control group and highest in the mRNA vaccine group, which had the fewest PFs but the highest number of ATFs. We hypothesize that this increased VEGF expression in both vaccine groups, especially the mRNA vaccine group, may be partly due to the elevated follicular TGF-β1 expression induced by the vaccines and partly as an adaptive response to support folliculogenesis and maintain ovarian function under altered microenvironmental conditions.

The cellular mechanism underlying ovarian follicular atresia and luteal regression is apoptosis [25]. Caspase-3 is the most critical executor of both endogenous and exogenous apoptotic pathways [43]. In rats, caspase-3 is localized in CLs, including luteal cells of healthy CLs and TCs [25]. Notably, no differences in caspase-3 staining intensity or distribution have been observed between healthy and apoptotic CLs [25]. In follicles, caspase-3 immunostaining is specific to GCs of ATFs and is absent in GCs of healthy follicles [25]. In a radiation-induced premature ovarian failure model, lower caspase-3 expression was observed in TCs of irradiated ovaries, with intense staining in growing preantral follicles and AFs [37]. Similarly, in a cisplatin-induced ovarian toxicity model, caspase-3 immunoreactivity was positive in oocytes and diffusely positive in stromal cells in the cisplatin-treated group [44]. Most GCs and TCs in the same study also exhibited positive caspase-3 staining, and Graafian follicles were present [44]. Another study demonstrated a higher number of caspase-3-positive cells in both the parenchymal and luteal structures of ovaries in the cisplatin group compared to controls [45]. In our study, caspase-3 expression was increased in both the mRNA and inactivated vaccine groups compared to controls, with more pronounced expression in oocytes, GCs, and interstitial stroma cells in the mRNA vaccine group. These findings align partly with those from experimental models of ovarian toxicity induced by cisplatin [44,45]. Consequently, our findings suggest that both mRNA and inactivated COVID-19 vaccines may induce apoptosis in oocytes, GCs, and interstitial stroma cells in rats through a caspase-dependent pathway, with a more pronounced effect observed following mRNA vaccine administration.

Various histomorphologic and biochemical methods, including AF and ATF counts and serum hormone concentrations, have been utilized to assess ovarian function and damage. Among these, AMH levels and AF counts are recognized as the most sensitive and specific indicators of ovarian reserve [46]. AMH expression is highest in the GCs of preantral follicles and small AFs, decreases in late AFs [47,48,49], and is absent in PrmFs and ATFs [49,50]. As a member of the TGF-β family, AMH inhibits initial follicular recruitment [47] and reduces follicular sensitivity to FSH during cyclic recruitment [48]. Studies in AMH-null female mice revealed that the absence of AMH leads to a significant depletion of PrmFs by 13 months of age [51]. When subjected to a superovulation protocol, these mice exhibited increased PrmF recruitment, which was counterbalanced by greater follicular loss during the transition from small preantral to large preantral follicles [52]. A recent meta-analysis [27] evaluating the effects of COVID-19 vaccines on assisted reproductive outcomes in infertile women included seven studies using mRNA and inactivated vaccines [9,53,54,55,56,57,58]. The meta-analysis found a significant increase in the proportion of mature oocytes obtained in vaccinated women [27]. Subgroup analysis showed that the number of oocytes retrieved was consistent across studies using mRNA vaccines, but the positive association between vaccination and the proportion of mature oocytes was not significant after excluding one study in the sensitivity analysis [27,56]. A recent meta-analysis assessing the impact of COVID-19 infection and vaccination on ovarian reserve, as measured by anti-Müllerian hormone (AMH) levels, included four studies investigating mRNA vaccines, one study evaluating the Russian GAM-COVID vaccine, and two studies in which the type of vaccine was not specified. The findings of this meta-analysis demonstrated a significant decline in AMH levels following SARS-CoV-2 infection in women of reproductive age, whereas COVID-19 vaccination was associated with a non-significant change in AMH levels [59]. However, it is important to note that the results were not consistent across all studies included in the meta-analysis [59].

Furthermore, elevated circulating follicle-stimulating hormone (FSH) levels have been shown to enhance follicular survival and enable antral follicle cohorts to evade apoptotic demise during cyclic recruitment [18]. Therefore, although the meta-analysis findings on oocyte yield in in vitro fertilization (IVF) cycles provide evidence that mRNA and inactivated COVID-19 vaccines do not negatively impact follicular development in IVF patients, it is essential to consider that FSH treatment was administered in these cycles to promote multifollicular development.

In our study, the mean numbers of PrmFs, PriFs, and SFs, as well as the intensity of AMH immunostaining in PriFs and SFs, were significantly lower in the inactivated vaccine group compared to the control group, with further reductions observed in the mRNA vaccine group. Moreover, in the mRNA vaccine group, the serum AMH level was reduced. These findings underscore the need for further investigation into the potential impacts of COVID-19 vaccination on ovarian reserve in clinical settings. Furthermore, this study focused on the effects of COVID-19 vaccination on ovarian follicles and ovarian reserve by evaluating TGF-β1, VEGF, caspase-3, and AMH expression, as well as follicular development stages in rat ovarian tissue, rather than assessing the immune response. However, previous studies have reported immune responses to vaccination. Research has shown that rats develop strong neutralizing antibody production, spike protein-specific T cell responses, and cytokine activation, comparable to human immune responses [60,61]. Additionally, preclinical evaluations of inactivated SARS-CoV-2 vaccines have demonstrated substantial antigen-specific immune activation in rodents, including increased interferon-gamma (IFN-γ) production and antibody titers similar to human data [62,63].

To ensure biologically relevant effects, ovarian tissue was collected four weeks after the second vaccine dose, aligning with peak immune activation observed in both humans and rodents [61,64]. Furthermore, inflammatory cytokines such as IL-6 and TGF-β1, which are commonly upregulated post-vaccination, were analyzed to enhance the translational relevance of the immunohistochemical findings [65]. CoronaVac (Sinovac Biotech Ltd., Beijing, China), an inactivated SARS-CoV-2 vaccine, was also tested in mice and rats during preclinical studies, providing essential safety and immunogenicity data while guiding dose selection [66]. Similarly, rats were used to determine initial immunogenicity data and optimize immunization doses and schedules for the BBIBP-CorV vaccine developed by the Chinese Center for Disease Control and Prevention and Beijing Institute of Biological Products Limited [63].

Although immune responses in rodents may not fully replicate those in humans [60,61], increasing cytokines such as IL-6, TNF-α, IL-1 β in mice [38] andTGF-β1 in human [36] and TGF-β1 play critical roles in follicular development, apoptosis, and ovarian remodeling [39]. Further research in both human and animal models is needed to clarify the relationship between COVID-19 vaccines, cytokines, and ovarian reserve.

This study was conducted in accordance with established guidelines [28,29,30], which recommend using a single dose capable of inducing an immune response in an animal model, matching the human dose without weight adjustment. Allometric scaling, a widely accepted approach in vaccine research, acknowledges that immune responses do not scale proportionally across species, making clinical dose-based assessments more appropriate than metabolic corrections. While allometric scaling suggests that administering a human dose to rats may result in a stronger immune response, adhering to the clinical dose is essential to avoid confounding effects from excessive immune activation [67].

The use of Wistar albino rats in this study is justified by their well-established role in reproductive toxicology and ovarian reserve research. Their short estrous cycle (4–5 days) allows for controlled monitoring of follicular development, atresia, and ovarian reserve changes [18]. Additionally, key ovarian processes such as follicular recruitment, selection, and atresia share fundamental similarities between rats and humans, making them a suitable model for reproductive toxicology assessments [13,15]. While folliculogenesis in humans spans several months and is primarily regulated by gonadotropins, in rats, follicular turnover is more rapid and less dependent on gonadotropins [68].

Our findings indicate that both mRNA and inactivated COVID-19 vaccines, particularly the mRNA vaccine, inhibit initial follicular recruitment and reduce AMH levels [47,48], leading to decreased follicular sensitivity to FSH during cyclic recruitment. Conversely, VEGF, which promotes granulosa cell proliferation and inhibits apoptosis [41], TGF-β1, which supports late-stage folliculogenesis by increasing angiogenic activity in granulosa cells [69], and caspase-3, a key regulator of apoptotic pathways [43], were all elevated following vaccination. Additionally, both vaccines led to a reduction in all follicular forms except atretic follicles (ATFs) compared to the control group. These findings suggest that the mRNA vaccine may accelerate follicular depletion by increasing recruitment and apoptosis, ultimately reducing the primordial follicle pool and ovarian reserve.

While this study provides insight into the potential effects of COVID-19 vaccines on ovarian function, its clinical relevance remains uncertain. Longitudinal studies assessing serum AMH levels and antral follicle counts in human populations are necessary to validate these preclinical findings and determine their significance in reproductive health.

Future research should incorporate allometric scaling principles to refine dose selection strategies. Establishing a dose–response relationship across different dose groups could enhance understanding of immune responses. Additionally, pharmacokinetic and pharmacodynamic analyses to evaluate antigen bioavailability and systemic distribution could further strengthen the clinical relevance of vaccine studies. These approaches may improve alignment between preclinical models and human immune responses, offering clearer insights into vaccine safety.

### Limitations

The current study has several limitations. Firstly, the study was conducted only on healthy rats; therefore, a comparison group of SARS-CoV-2-infected rats is needed to analyze cellular immune responses (e.g., CD4+ and CD8+ T cells, Tregs, and cytokines) comprehensively. Such a comparison would clarify the immunological impact of COVID-19 vaccines on ovarian tissue. Secondly, hormonal analyses, including fluctuations in estradiol, FSH, and LH levels, were not performed. These evaluations would strengthen the understanding of the vaccine’s effects on ovarian follicles. Thirdly, the study focused solely on the short-term impacts of COVID-19 vaccines on ovarian tissue. Evaluating long-term effects, including fertility outcomes, offspring viability, and inflammatory responses, would provide more extensive insights. Future studies should incorporate immunohistochemistry or flow cytometry to examine immune cell infiltration in ovarian tissue, comparing vaccinated and unvaccinated groups to determine immune-mediated mechanisms underlying observed changes. Additionally, assessing long-term fertility and offspring health could clarify whether vaccine-induced ovarian alterations affect reproductive function or if compensatory mechanisms exist.

## 5. Conclusions

This study is the first to evaluate the effects of mRNA and inactivated COVID-19 vaccines on ovarian follicles and ovarian reserve in a rat model. The findings demonstrate that both vaccines, particularly the mRNA vaccine, are associated with a reduction in ovarian reserve, characterized by depletion of the primordial follicle pool and increased follicular loss via apoptosis throughout folliculogenesis. However, these results should be interpreted cautiously, as preclinical models cannot be directly extrapolated to human reproductive health. Further longitudinal studies in human populations, including assessments of AMH levels and antral follicle counts, are necessary to validate these observations and determine their clinical significance.

## Figures and Tables

**Figure 1 vaccines-13-00345-f001:**
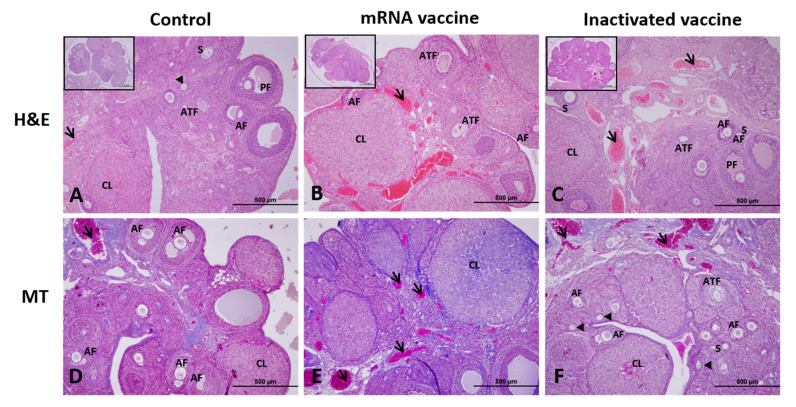
Ovarian images of the control, mRNA vaccine, and inactivated vaccine groups after Hematoxylin and Eosin (H&E; **A**–**C**), and Masson’s Trichrome stainings (MT; **D**–**F**). CL; corpus luteum, ATF; atretic follicle, S; secondary follicle, AF; antral follicle, PF: preovulatory follicles, Arrow; blood vessel, Arrowhead; primary follicle. Scale bars: 500 µm.

**Figure 2 vaccines-13-00345-f002:**
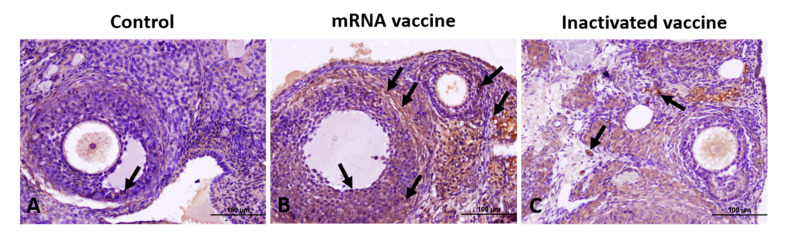
Immunohistochemical staining of TGF-β1 in ovarian sections from the control (**A**), mRNA vaccine (**B**), and inactivated vaccine (**C**) groups. Arrows: immunopositive cells. Scale bars: 100 µm.

**Figure 3 vaccines-13-00345-f003:**
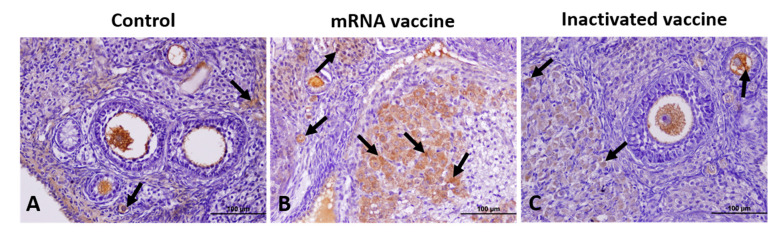
Immunohistochemical staining of VEGF in ovarian sections from the control (**A**), mRNA vaccine (**B**), and inactivated vaccine (**C**) groups. Arrows: immunopositive cells. Scale bars: 100 µm.

**Figure 4 vaccines-13-00345-f004:**
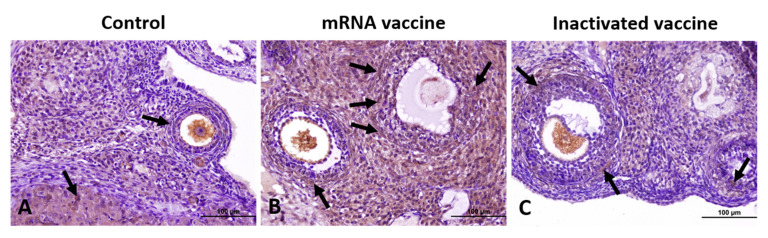
Immunohistochemical staining of Caspase-3 in ovarian sections from the control (**A**), mRNA vaccine (**B**), and inactivated vaccine (**C**) groups. Arrows: immunopositive cells. Scale bars: 100 µm.

**Figure 5 vaccines-13-00345-f005:**
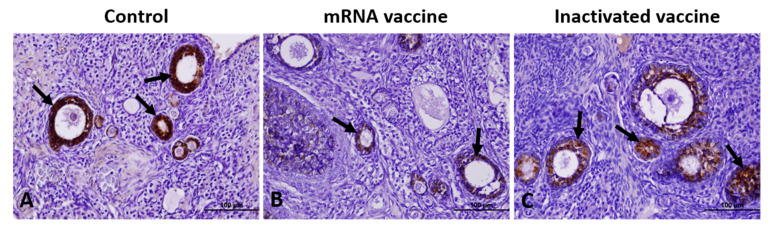
Immunohistochemical staining of AMH in ovarian sections from the control (**A**), mRNA vaccine (**B**), and inactivated vaccine (**C**) groups. Arrows: immunopositive cells. Scale bars: 100 µm.

**Figure 6 vaccines-13-00345-f006:**
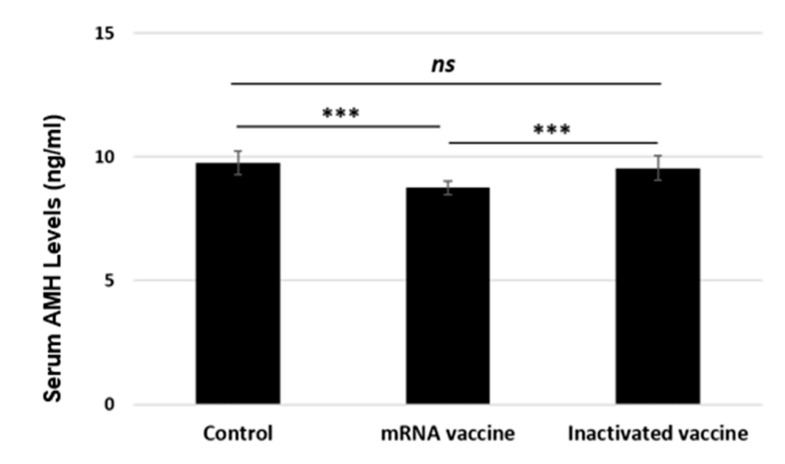
Comparison of serum AMH levels in the control, mRNA vaccine, and inactivated vaccine groups.ns; not significant, *** *p* < 0.001.

**Table 1 vaccines-13-00345-t001:** Comparison of follicle numbers in the control, mRNA vaccine, and inactivated vaccine groups.

Parameter	Control Group	mRNA Vaccine Group	Inactivated Vaccine Group	*p*
Number of Follicles				
Primordial	106.70 ± 5.33 ^a^	42.40 ± 4.96 ^b^	70.10 ± 12.04 ^c^	0.001
Primary	45.20 ± 4.21 ^a^	29.40 ± 4.57 ^b^	38.30 ± 5.41 ^c^	0.001
Secondary	39.60 ± 6.09 ^a^	25.30 ± 2.86 ^b^	32.70 ± 8.07 ^c^	0.001
Antral	32.80 ± 5.59 ^a^	26.20 ± 3.01 ^b^	29.40 ± 4.30 ^ab^	0.010
Preovulatory	18.20 ± 1.87 ^a^	15.00 ± 3.12 ^b^	16.20 ± 3.04 ^ab^	0.045
Atretic	13.90 ± 2.07 ^a^	26.40 ± 5.06 ^b^	17.10 ± 1.96 ^a^	0.001

Data are expressed as mean ± standard deviation. The same letters in the same row indicate a similarity between groups and different letters indicate a difference between groups.

**Table 2 vaccines-13-00345-t002:** Immunostaining intensities of follicular AMH, TGF-β1, VEGF, and caspase-3 in the control, mRNA vaccine, and inactivated vaccine groups.

Parameter	Control Group	mRNA Vaccine Group	Inactivated Vaccine Group	*p*
Follicular AMH				
Primary	129.09 ± 10.30 ^a^	100.35 ± 20.88 ^b^	111.43 ± 12.06 ^c^	0.001
Secondary	119.23 ± 10.09 ^a^	97.91 ± 9.71 ^b^	106.20 ± 13.67 ^c^	0.001
Antral	104.77 ± 14.00 ^a^	88.68 ± 10.38 ^b^	94.70 ± 11.31 ^b^	0.001
**TGF-β1**	85.08 ± 7.87 ^a^	130.78 ± 14.58 ^b^	120.63 ± 10.30 ^c^	0.001
**VEGF**	91.40 ± 5.57 ^a^	145.15 ± 7.36 ^b^	113.46 ± 5.57 ^c^	0.001
**Caspase-3**	82.78 ± 6.80 ^a^	138.09 ± 7.94 ^b^	98.85 ± 8.69 ^c^	0.001

Data are expressed as mean ± standard deviation. The same letters in the same row indicate a similarity between groups and different letters indicate a difference between groups.

## Data Availability

The data generated in the present study may be requested from the corresponding author.

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
