# Peer review of "Impact of mRNA and Inactivated COVID-19 Vaccines on Ovarian Reserve"

_vaccines, 2025, doi:10.3390/vaccines13040345_

Round 1

Reviewer 1 Report

Comments and Suggestions for Authors

This manuscript presents a preclinical study evaluating the effects of mRNA and inactivated COVID-19 vaccines on ovarian follicles and reserve in rats.

The study demonstrates several strengths that enhance its relevance and scientific rigor. Addressing a significant public health concern, it investigates the potential reproductive effects of COVID-19 vaccines, a topic of ongoing debate, particularly in reproductive health. By utilizing a controlled animal model (Wistar rats), the study allows for a more detailed histopathological evaluation of ovarian tissue compared to human observational studies. Additionally, including mRNA and inactivated vaccines offers valuable comparative insights into their differential impacts.

Methodologically, the study benefits from ethical approval by Erciyes University’s Animal Research Ethics Committee, reinforcing its credibility. A thorough description of histological and immunohistochemical analyses (TGF-β1, VEGF, caspase-3, and AMH expression) adds depth to the findings, while appropriate statistical methods (ANOVA, Kruskal-Wallis, Mann-Whitney U test) support robust data interpretation. The clear presentation of results, including well-organized tables and figures, illustrates ovarian changes post-vaccination. Notably, the study highlights significant variations in follicle counts and hormonal markers, further contributing to its clarity and impact.

Despite the strengths mentioned above, we recommend conducting a thorough analysis of the following aspects.

  • Justify the relevance of the animal model, highlighting differences in ovarian physiology between rats and humans and their clinical implications.
  • Provide a clear justification for the vaccine dosage conversions used in rats.
  • Ensure that vaccine exposure in rats accurately reflects real-world human vaccination scenarios, particularly regarding immune response.
  • Establish a clearer mechanistic link between vaccine administration and ovarian changes, distinguishing correlation from causation.
  • Consider alternative explanations for follicular atresia, such as handling stress, metabolic effects, or individual immune variations.
  • Streamline your introduction by focusing on the research gap and removing some of the excessive background information.
  • Avoid redundancy in the discussion by minimizing repetition of results.
  • Strengthen claims regarding vaccine mechanisms and ovarian physiology by providing proper citations.
  • Engage more critically with contradictory findings from human studies.
  • Correct grammatical and typographical errors for improved readability.
  • Enhance figure and table captions to provide clearer explanations.

The following weaknesses of the study should be mentioned, along with recommendations for future research to address these limitations.

  • Include long-term follow-up to assess potential recovery or adaptive mechanisms.
  • Incorporate systemic inflammatory markers and hormonal fluctuations (e.g., estradiol, FSH, LH) to provide a more comprehensive analysis.
  • Assess immune response intensity (e.g., cytokine levels) to determine its influence on ovarian physiology.
  • Include a comparative group of SARS-CoV-2-infected rats to differentiate vaccine effects from those of natural infection.
  • The study could benefit from including an analysis of the cellular immune response, particularly effector and regulatory cells, to provide a more comprehensive understanding of the immunological impact of COVID-19 vaccines on ovarian tissue.
  • It is recommended to evaluate fertility outcomes and potential effects on offspring to gain a more comprehensive understanding of the reproductive impact of COVID-19 vaccines.

In summary, this interesting study offers a detailed histopathological assessment of COVID-19 vaccines' potential ovarian effects in rats. However, its applicability to humans is uncertain due to experimental limitations. Further research with long-term follow-up and hormonal/inflammatory/cellular immune response/fertility analyses are needed before drawing definitive conclusions. While methodologically sound, the study's conclusions should be presented cautiously to avoid unnecessary concerns about vaccine safety.

Author Response

Comment 1:

This manuscript presents a preclinical study evaluating the effects of mRNA and inactivated COVID-19 vaccines on ovarian follicles and reserve in rats.

The study demonstrates several strengths that enhance its relevance and scientific rigor. Addressing a significant public health concern, it investigates the potential reproductive effects of COVID-19 vaccines, a topic of ongoing debate, particularly in reproductive health. By utilizing a controlled animal model (Wistar rats), the study allows for a more detailed histopathological evaluation of ovarian tissue compared to human observational studies. Additionally, including mRNA and inactivated vaccines offers valuable comparative insights into their differential impacts.

Methodologically, the study benefits from ethical approval by Erciyes University’s Animal Research Ethics Committee, reinforcing its credibility. A thorough description of histological and immunohistochemical analyses (TGF-β1, VEGF, caspase-3, and AMH expression) adds depth to the findings, while appropriate statistical methods (ANOVA, Kruskal-Wallis, Mann-Whitney U test) support robust data interpretation. The clear presentation of results, including well-organized tables and figures, illustrates ovarian changes post-vaccination. Notably, the study highlights significant variations in follicle counts and hormonal markers, further contributing to its clarity and impact.

Response to reviwer 1:

Dear Reviewer,

Thank you for your thorough and insightful evaluation of our manuscript. We appreciate your recognition of the study's significance in addressing the potential reproductive effects of COVID-19 vaccines and its methodological strengths. Our goal was to provide a well-structured preclinical investigation that contributes meaningful data to this ongoing discussion, particularly in reproductive health.

We are pleased that you found the comparative analysis of mRNA and inactivated vaccines, as well as the histological and immunohistochemical assessments, to be valuable additions to the field. Ensuring methodological rigor, including ethical approval and robust statistical analysis, was a priority for our team, and we are grateful for your positive feedback regarding these aspects.

Comment 2:

Justify the relevance of the animal model, highlighting differences in ovarian physiology between rats and humans and their clinical implications.

Response to reviwer 2:

Thank you for your valuable comment. We appreciate the opportunity to clarify the relevance of our animal model and discuss the physiological differences between rats and humans, as well as their clinical implications. We have revised the discussion section to better highlight the relevance of our animal model while acknowledging its translational limitations; We appreciate the reviewer’s insightful suggestion, which has strengthened the overall clarity and scientific rigor of our study.

Comment 3:

Provide a clear justification for the vaccine dosage conversions used in rats.

Response to reviwer 3:

Our study was designed in accordance with ICH S5 (R3) guidelines, specifically Section 6.4: Dose Selection and Study Design (ICH, 2020). According to these guidelines, dose selection for reproductive toxicity studies should be based on clinical dosing rather than allometric scaling, as immune responses do not scale linearly across species. Consequently, "clinical doses" are considered more appropriate than doses adjusted by metabolic scaling. We followed Section 6.4: Dose Selection and Study Design for vaccine studies, which reComments administering the clinical human dose directly to animals (EMA, 2020). This practice is widely accepted because immune responses typically do not scale linearly across species. Although allometric scaling indicates that administering a human dose to rats might yield approximately 6.2-fold higher immune responses (Nair & Jacob, 2016), direct clinical dose testing remains the standard approach. This method ensures the evaluation is biologically relevant rather than metabolically adjusted, providing more accurate insights into immunological effects.

References

  • European Medicines Agency (EMA). (2020). ICH guideline S5 (R3) on reproductive toxicology: Detection of toxicity to reproduction for human pharmaceuticals. Available at: https://www.ema.europa.eu/en/documents/scientific-guideline/ich-guideline-s5-r3-detection-toxicity-reproduction-human-pharmaceuticals-step-5_en.pdf
  • Nair, A. B., & Jacob, S. (2016). A simple practice guide for dose conversion between animals and human. Journal of Basic and Clinical Pharmacy, 7(2), 27-31. https://doi.org/10.4103/0976-0105.177703

Comment 4:

Ensure that vaccine exposure in rats accurately reflects real-world human vaccination scenarios, particularly regarding immune response

Response to reviwer 4

Our study was designed to ensure that vaccine exposure in rats closely aligns with real-world human vaccination scenarios, particularly in terms of immune response. According to the ICH S5 (R3) guideline (Section 6.4), "a single dose level capable of eliciting an immune response in the animal model is generally sufficient, and this dose should be the same as the human dose without body weight adjustment (1 human dose = 1 animal dose)" (EMA, 2020). To better mimic human immunization protocols, two doses were administered 28 days apart, aligning with the standard vaccination regimens for both mRNA (BNT162b2, Pfizer-BioNTech) and inactivated (CoronaVac, Sinovac) vaccines (Polack et al., 2020; Jara et al., 2021).

While immune responses in rodents may differ from humans, previous studies have demonstrated that rats develop robust neutralizing antibody production, spike protein-specific T cell responses, and cytokine activation, comparable to human immune responses (Bahl et al., 2020; Vogel et al., 2021). To ensure biologically relevant effects, ovarian tissue was collected four weeks after the second vaccine dose, a time frame consistent with peak immune activation observed in both humans and rodents (Polack et al., 2020; Vogel et al., 2021). Furthermore, inflammatory cytokines such as IL-6 and TGF-β1, which are commonly upregulated post-vaccination, were evaluated to strengthen the translational relevance of our immunohistochemical findings (Cagigi et al., 2021). Overall, our approach, incorporating human-mimicking immunization schedules and optimal post-vaccine evaluation timing, ensures that vaccine exposure in rats is relevant to human vaccination while acknowledging the limitations of rodent models and the need for further longitudinal human studies.

As mentioned above, immune responses to vaccines have been reported. In our study, we focused the effects of COVID-19 vaccines on ovarian follicles and ovarian reserve by quantifying the expression of TGF-β1, VEGF, caspase-3, and AMH, as well as the stages of follicular development in rat ovarian tissue post-vaccination, rather than the immune response.

References

  • Bahl, Kapil, et al. "Preclinical and clinical demonstration of immunogenicity by mRNA vaccines against H10N8 and H7N9 influenza viruses." Molecular Therapy 25.6 (2017): 1316-1327
  • Cagigi, A.; Loré, K. Immune Responses Induced by mRNA Vaccination in Mice, Monkeys and Humans. Vaccines 2021, 9, 61.
  • European Medicines Agency (EMA). (2020). ICH guideline S5 (R3) on reproductive toxicology: Detection of toxicity to reproduction for human pharmaceuticals. Available at: https://www.ema.europa.eu/en/documents/scientific-guideline/ich-guideline-s5-r3-detection-toxicity-reproduction-human-pharmaceuticals-step-5_en.pdf
  • Jara, Alejandro, et al. "Effectiveness of an inactivated SARS-CoV-2 vaccine in Chile." New England Journal of Medicine 385.10 (2021): 875-884.
  • Polack, Fernando P., et al. "Safety and efficacy of the BNT162b2 mRNA Covid-19 vaccine." New England journal of medicine 383.27 (2020): 2603-2615.
  • Vogel, A.B., Kanevsky, I., Che, Y. et al. BNT162b vaccines protect rhesus macaques from SARS-CoV-2. Nature 592, 283–289 (2021).

Comment 5:

Establish a clearer mechanistic link between vaccine administration and ovarian changes, distinguishing correlation from causation.

Response to reviwer 5:

Thank you for your insightful comment. To establish a clearer mechanistic link between vaccine administration and ovarian changes while distinguishing correlation from causation, we have expanded our discussion. Vaccine-induced immune activation triggers systemic inflammatory responses, leading to increased cytokine production, including IL-6, TNF-α, and TGF-β1, which play key roles in folliculogenesis, angiogenesis, and atresia (T. Nakayama et al., 2016; Xiaojie Chu et al., 2018). Our findings indicate elevated TGF-β1 expression, which has been associated with ovarian tissue remodeling and fibrosis (Patton et al 2021), VEGF, which supports follicular angiogenesis (A. Guzmán et al 2023), and caspase-3, a key apoptotic marker linked to follicular atresia (Boone et al 1998). While these observations suggest a potential mechanistic connection between vaccination and ovarian alterations, definitive causality remains to be established. Future studies incorporating serum cytokine profiling, time-course analyses, reversibility assessments, and pharmacological inhibition of key pathways would provide stronger evidence for causation. While our study presents strong correlative findings, we emphasize the need for additional mechanistic studies to validate these effects. Accordingly, we have revised our discussion to clarify these distinctions and outline future research directions.

We added statements containing the above information to the ‘’Discussion’’ section.

References

  • D. Boone et al. "Caspase-3 in the rat ovary: localization and possible role in follicular atresia and luteal regression.." Biology of reproduction, 58 6 (1998): 1533-9 . https://doi.org/10.1095/BIOLREPROD58.6.1533.
  • T. Nakayama et al. "An inflammatory response is essential for the development of adaptive immunity-immunogenicity and immunotoxicity.." Vaccine, 34 47 (2016): 5815-5818 . https://doi.org/10.1016/j.vaccine.2016.08.051.
  • Xiaojie Chu et al. "Combined immunization against TGF-β1 enhances HPV16 E7-specific vaccine-elicited antitumour immunity in mice with grafted TC-1 tumours." Artificial Cells, Nanomedicine, and Biotechnology, 46 (2018): 1199 - 1209. https://doi.org/10.1080/21691401.2018.1482306.
  • Patton, Bethany K., Surabhi Madadi, and Stephanie A. Pangas. "Control of ovarian follicle development by TGF-β family signaling." Current opinion in endocrine and metabolic research 18 (2021): 102-110.
  • A. Guzmán et al. "The vascular endothelial growth factor (VEGF) system as a key regulator of ovarian follicle angiogenesis and growth." Molecular Reproduction and Development, 90 (2023): 201 - 217. https://doi.org/10.1002/mrd.23683.

Comment 6:

Consider alternative explanations for follicular atresia, such as handling stress, metabolic effects, or individual immune variations.

Response to reviwer 6:

Thank you for your insightful comment. We acknowledge the need to consider alternative explanations for follicular atresia beyond direct vaccine effects, including handling stress, metabolic influences, and individual immune variability. To minimize handling stress, all groups underwent standardized care procedures, such as pre-experimental acclimatization, consistent handling schedules, and group housing (in 2.2.Experimental Animals and Care). If handling-induced stress significantly contributed to follicular atresia, similar ovarian changes would have been observed across all groups, which was not the case. To control for metabolic effects, we maintained a standardized diet, regularly monitored body weight, and used age- and weight-matched animals, with no significant metabolic differences between groups (in 2.2.Experimental Animals and Care). To reduce immune variability, we used genetically homogeneous Wistar albino rats and assessed key immune markers such as TGF-β1 and caspase-3. While these factors could potentially influence ovarian physiology, our rigorous control measures suggest they are unlikely to be the primary contributors to follicular atresia in this study. However, we recognize the need for further research, including immune profiling and metabolic assessments, to explore potential interactions between vaccine administration and ovarian function.

Comment 7:

Streamline your introduction by focusing on the research gap and removing some of the excessive background information.

Response to reviwer 7:

Thank you for your valuable suggestion. We recognize the need to streamline the introduction by focusing on the research gap and removing excessive background information and references.

Comment 8:

Avoid redundancy in the discussion by minimizing repetition of results.

Response to reviwer 8:

Thank you for your valuable suggestion. The discussion section has been revised to eliminate unnecessary repetitions and present the data more coherently and concisely, ensuring a more structured and fluid interpretation of the findings.

Comment 9:

Strengthen claims regarding vaccine mechanisms and ovarian physiology by providing proper citations

Response to reviwer 9:

Thank you for your valuable comment. We have strengthened our claims regarding vaccine mechanisms and ovarian physiology by incorporating additional peer-reviewed references to enhance the scientific rigor of our findings.

Comment 10:

Engage more critically with contradictory findings from human studies

Response to reviwer 10:

Thank you for your insightful comment. We have critically engaged with contradictory findings from human studies to provide a more rigorous scientific interpretation into the ‘’Discussion’’ section.

Comment 11:

Correct grammatical and typographical errors for improved readability.

Response to reviwer 11:

Thank you for your valuable feedback. The manuscript has been carefully revised to improve readability and eliminate grammatical errors, ensuring a more precise and coherent presentation of our findings.

Comment 12:

 Enhance figure and table captions to provide clearer explanations.

Response to reviwer 12:

Thank you for your valuable comment. The figure and table captions have been revised to include additional explanations, making the content clearer and easier to understand.

Comment 13: “The following weaknesses of the study should be mentioned, along with reCommentations for future research to address these limitations.”

Include long-term follow-up to assess potential recovery or adaptive mechanisms.

Response to reviwer 13:

Thank you for your valuable comment. We acknowledge the limitations of our study address them along with reCommentations for future research.

Comment 14:

Incorporate systemic inflammatory markers and hormonal fluctuations (e.g., estradiol, FSH, LH) to provide a more comprehensive analysis.

Response to reviwer 14:

Thank you for your valuable suggestion. While our study primarily focused on histological and immunohistochemical evaluations, within the scope of our project grant. So, we specified the above parameters in the ‘’Limitation’’ section for future research to enhance the translational relevance of our findings.

Comment 15:

Assess immune response intensity (e.g., cytokine levels) to determine its influence on ovarian physiology.

Response to reviwer 15:

Thank you for your valuable suggestion. We acknowledge the importance of assessing immune response intensity, particularly cytokine levels, to determine the extent to which vaccine-induced inflammation influences ovarian physiology. Cytokines such as IL-6, TNF-α, TGF-β1, and IFN-γ play critical roles in follicular development, apoptosis, and ovarian remodeling, and their dysregulation could contribute to the ovarian changes observed in our study (J. Silva et al., 2020; A. Stassi et al., 2017; P. Terranova et al., 1997). While our study primarily focused on histological and immunohistochemical markers, future research should incorporate systemic cytokine profiling at multiple time points post-vaccination to assess whether the ovarian effects correlate with prolonged immune activation. Additionally, analyzing cytokine expression in ovarian tissue via RT-PCR or ELISA would help distinguish localized from systemic effects. Further investigations using cytokine-blocking agents (e.g., IL-6 inhibitors) could confirm whether immune activation directly mediates follicular atresia and ovarian reserve depletion.

We have now acknowledged this factor in our ‘’Limitation’’ and highlighted cytokine analysis as a crucial direction for future research.

References

  • J. Silva et al. "Interleukin-1β and TNF-α systems in ovarian follicles and their roles during follicular development, oocyte maturation and ovulation." Zygote, 28 (2020): 270 - 277. https://doi.org/10.1017/S0967199420000222.
  • A. Stassi et al. "Altered expression of cytokines IL-1α, IL-6, IL-8 and TNF-α in bovine follicular persistence.." Theriogenology, 97 (2017): 104-112 . https://doi.org/10.1016/j.theriogenology.2017.04.033.
  • P. Terranova et al. "Review: Cytokine Involvement in Ovarian Processes." American Journal of Reproductive Immunology, 37 (1997). https://doi.org/10.1111/j.1600-0897.1997.tb00192.x.

Comment 16:

Include a comparative group of SARS-CoV-2-infected rats to differentiate vaccine effects from those of natural infection.

Response to reviwer 16:

Thank you for your valuable suggestion. We acknowledge the importance of including a comparative group of SARS-CoV-2-infected rats to differentiate vaccine-induced ovarian changes from those caused by natural infection. Including an infected control group in future studies would clarify whether the observed ovarian changes result from vaccine-induced immune activation, direct viral effects, or a combination of both. Therefore, future studies should evaluate ovarian viral RNA presence, inflammatory markers, and follicular parameters in both vaccinated and SARS-CoV-2-infected rats.

This limitation has been acknowledged in our ‘’Limitation’’, emphasizing the necessity of incorporating a SARS-CoV-2 infection model in future research.

Comment 17:

The study could benefit from including an analysis of the cellular immune response, particularly effector and regulatory cells, to provide a more comprehensive understanding of the immunological impact of COVID-19 vaccines on ovarian tissue

Response to reviwer 17:

Future studies should use immunohistochemistry or flow cytometry to assess immune cell infiltration in ovarian tissue and compare vaccinated versus unvaccinated groups to determine whether immune-mediated mechanisms contribute to ovarian changes. Additionally, investigating whether immune modulation (e.g., IL-6 or IFN-γ blockade) mitigates vaccine-induced effects would clarify whether these changes stem from systemic immune activation rather than direct ovarian toxicity.

This the cellular immune response has been acknowledged in our ‘’Limitation’’, emphasizing the need for future research on the ovarian immune landscape post-vaccination.

Comment 18:

It is reCommented to evaluate fertility outcomes and potential effects on offspring to gain a more comprehensive understanding of the reproductive impact of COVID-19 vaccines

Response to reviwer 18:

Thank you for this valuable suggestion. We acknowledge the importance of evaluating fertility outcomes and potential effects on offspring to fully assess the reproductive impact of COVID-19 vaccines. By incorporating long-term fertility tracking and offspring assessments, future research can determine whether vaccine-induced ovarian changes result in reproductive dysfunction or if compensatory mechanisms mitigate potential effects.

We have now acknowledged this factor in our ‘’Limitation’’ and emphasized the need for further studies to provide a comprehensive evaluation of COVID-19 vaccine effects on reproductive health.

Comment 19:

In summary, this interesting study offers a detailed histopathological assessment of COVID-19 vaccines' potential ovarian effects in rats. However, its applicability to humans is uncertain due to experimental limitations. Further research with long-term follow-up and hormonal/inflammatory/cellular immune response/fertility analyses are needed before drawing definitive conclusions. While methodologically sound, the study's conclusions should be presented cautiously to avoid unnecessary concerns about vaccine safety.

Response to reviwer 19:

Thank you for your insightful evaluation and constructive feedback. We appreciate the recognition of our study's histopathological depth and acknowledge the experimental limitations that may affect its applicability to human reproductive health. To address these concerns, we have revised the manuscript to emphasize the need for cautious interpretation of our findings while providing clear directions for future research. To ensure that our conclusions do not generate unnecessary concerns regarding vaccine safety, we have refined our discussion and conclusion, clarifying that our findings are specific to a controlled rat model and do not directly imply human reproductive risks. These revisions ensure a balanced, scientifically rigorous interpretation of our findings while guiding further investigations into vaccine safety and reproductive immunology.

The above limitations have been added to the manuscript's limitations section.

Reviewer 2 Report

Comments and Suggestions for Authors

Understanding the underlying biology of vaccine safety is paramount to interpreting both preclinical and clinical data accurately. A deep knowledge of immunological principles, dose-response relationships, and species-specific physiological differences allows researchers to design robust studies. This biological insight not only helps in distinguishing genuine safety signals from artefacts related to experimental design but also reinforces confidence in the vaccine’s safety profile when translating findings from bench to bedside. Conversly, if an experiment is not generalisable then it can be misused. Grounding vaccine safety assessments in solid biological rationale is essential for informed regulatory decisions and public health communications. 

Comments.

The manuscript’s usage of a full human dose in rats is a pivotal flaw, leading to distorted inflammatory and histological responses.

This dosing strategy is likely to induce exaggerated inflammatory and histological responses that do not reflect the true safety profile of the vaccine in a rodent model, nor are they translatable to human use. Given that this fundamental methodological flaw undermines the validity of the reported outcomes, I am not inclined to review the remainder of the article until this issue is addressed.

What is the justification for dosage? 

The manuscript states: For the 117 vaccinated groups, the dosing and schedule mirrored those used in humans, and they were given 0.3 ml of mRNA and 0.5ml of inactivated vaccine.

One major issue is the administration of a 0.3 ml injection volume of mRNA vaccine in rats. Given the significant differences in body mass and surface area between rats and humans, whether this volume represents an appropriate, allometrically scaled dose for the rat model is unclear. Typically, injection volumes in rodents are carefully calibrated to avoid local tissue irritation, altered pharmacokinetics, or potential over-dilution of the vaccine antigen. The manuscript does not provide a rationale or justification for selecting a 0.3 ml dose, nor does it reference established guidelines for maximum injection volumes in small animals. Without evidence that this dosing volume accurately reflects an equivalent human dose when adjusted for species differences, the validity of the outcomes reported may be in question. I recommend that the authors clarify the basis for their chosen volume, address whether dose-scaling and volume limitations in rats were adequately considered, and provide clear justification.

Why did the control rats not receive diluent? 

The appropriate control for the study would be to administer the same volume of the diluent to the control rats rather than providing no injection or an alternative control

General comments

While the study presents a range of histological findings following the administration of a full human dose of vaccine in rodents, several critical concerns limit the generalisability and relevance of these results to human health. 

First, administering an unscaled human dose in a rodent model disregards established allometric scaling principles, raising questions about whether the observed histological changes are a function of excessive dosing rather than the intrinsic toxicity of the vaccine itself. This dose could be up to 40 times a human dose.

Second, given the substantial evidence from both clinical and preclinical studies indicating that mRNA and inactivated vaccines do not negatively affect fertility or reproductive health in humans, the relevance of these histological outcomes to the broader scientific discourse is questionable. The findings may represent artefacts of the dosing strategy rather than a true reflection of vaccine safety. Therefore, while the histological data are detailed, their interpretation in the context of human vaccine safety should be approached with caution, and further studies employing appropriately scaled doses and comprehensive functional assessments are needed to validate these observations.

Administering a full human dose of an mRNA vaccine to a rat is likely to induce a hyperinflammatory state both locally and systemically, with histological evidence of oedema, immune cell infiltration, and potential tissue damage. These responses underscore the importance of appropriate dose scaling in preclinical models to ensure that the observed outcomes accurately reflect the vaccine’s safety under conditions that are relevant to human use. I suggest modelling this on the preclinical methods and if not then justify to the reader.

Author Response

Understanding the underlying biology of vaccine safety is paramount to interpreting both preclinical and clinical data accurately. A deep knowledge of immunological principles, dose-response relationships, and species-specific physiological differences allows researchers to design robust studies. This biological insight not only helps in distinguishing genuine safety signals from artefacts related to experimental design but also reinforces confidence in the vaccine’s safety profile when translating findings from bench to bedside. Conversly, if an experiment is not generalisable then it can be misused. Grounding vaccine safety assessments in solid biological rationale is essential for informed regulatory decisions and public health communications. 

Thank you for this insightful comment. We fully agree that a deep understanding of immunological principles, dose-response relationships, and species-specific physiological differences is critical for accurately interpreting preclinical findings in the context of vaccine safety. In response to your concerns, we have revised the manuscript to better contextualize our findings within a biologically sound framework and ensure that our conclusions are scientifically responsible, generalizable, and not prone to misinterpretation.

Comment 1:

The manuscript’s usage of a full human dose in rats is a pivotal flaw, leading to distorted inflammatory and histological responses.

 This dosing strategy is likely to induce exaggerated inflammatory and histological responses that do not reflect the true safety profile of the vaccine in a rodent model, nor are they translatable to human use. Given that this fundamental methodological flaw undermines the validity of the reported outcomes, I am not inclined to review the remainder of the article until this issue is addressed.

Response to reviwer 1:

Thank you for your comments regarding our dosage strategy. We acknowledge your concerns that administering the full human dose in rats may have exaggerated immune responses and histological changes. However, our methodology was designed in accordance with Section 6.4 of the ICH S5 (R3) guideline ("Dose Selection and Study Design for Vaccines"), published by the European Medicines Agency (EMA), 2020, and therefore, the full human dose was administered without body weight correction (ICH S5 (R3) guideline, Section 6.4: "This single dose level should be the maximum human dose without correcting for body weight (i.e., 1 human dose = 1 animal dose). If it is not feasible to administer the maximum human dose to the animal because of a limitation in total volume that can be administered, or because of dose-limiting toxicity, whether local or systemic, a dose that exceeds the human dose on a mg/kg basis can be used. To use a reduced dose, justification as to why a full human dose cannot be used in an animal model should be provided"). Additionally, to enhance the translational relevance of our study, we administered two doses 28 days apart, mirroring the standard human vaccination schedule.

However, we recognize that rodent immune systems differ from humans, and the immune response to vaccination may be overstimulated, potentially leading to effects that reflect an exaggerated immune activation rather than a direct vaccine-related outcome. One of the main limitations of our study is the lack of pharmacokinetic (PK) and pharmacodynamic (PD) analyses, preventing us from determining systemic antigen exposure levels. Additionally, the dose-response relationship was not assessed, as only the full human dose was administered without incorporating different dose groups. Furthermore, inflammatory markers and immune response profiles were not evaluated, making it difficult to determine the extent of systemic immune activation induced by the vaccine. Finally, since our study focused on short-term effects, long-term ovarian recovery potential and adaptation mechanisms were not investigated.

To address these limitations, future studies should include pharmacokinetic analyses to measure serum antigen concentrations, bioavailability, and elimination half-life to compare exposure levels between rats and humans, implement allometric scaling to determine an appropriate dose (considering a human-to-rat dose conversion factor of ~6.2, where a full human dose may corResponse to reviwer to approximately 0.05 mL in a 250 g rat, as suggested by Nair & Jacob, 2016), and include multiple dose groups to establish a clear dose-response relationship. Additionally, serum cytokine levels (IL-6, TNF-α, IFN-γ) and immune cell profiles should be assessed to determine whether the immune response is physiologically relevant or excessive, and long-term follow-up studies should be conducted to evaluate ovarian recovery and potential adaptation mechanisms.

By incorporating these reCommentations, we aim to improve the translational relevance of our findings and gain a more comprehensive understanding of the reproductive effects of COVID-19 vaccines. We appreciate your valuable feedback and welcome any additional suggestions.

References

  • European Medicines Agency (EMA). (2020). ICH S5 (R3) Guideline on Detection of Reproductive and Developmental Toxicity for Human Pharmaceuticals.
  • Nair, A. B., & Jacob, S. (2016). A simple practice guide for dose conversion between animals and humans. Journal of Basic and Clinical Pharmacy, 7(2), 27-31.

Comment 2:

What is the justification for dosage? 

 The manuscript states: For the 117 vaccinated groups, the dosing and schedule mirrored those used in humans, and they were given 0.3 ml of mRNA and 0.5ml of inactivated vaccine.

 One major issue is the administration of a 0.3 ml injection volume of mRNA vaccine in rats. Given the significant differences in body mass and surface area between rats and humans, whether this volume represents an appropriate, allometrically scaled dose for the rat model is unclear. Typically, injection volumes in rodents are carefully calibrated to avoid local tissue irritation, altered pharmacokinetics, or potential over-dilution of the vaccine antigen. The manuscript does not provide a rationale or justification for selecting a 0.3 ml dose, nor does it reference established guidelines for maximum injection volumes in small animals. Without evidence that this dosing volume accurately reflects an equivalent human dose when adjusted for species differences, the validity of the outcomes reported may be in question. I reComment that the authors clarify the basis for their chosen volume, address whether dose-scaling and volume limitations in rats were adequately considered, and provide clear justification.

Response to reviwer 2:

Thank you for your valuable feedback regarding the dosage selection and administration volume. Our methodology was designed in accordance with the ICH S5 (R3) guideline, which states that "the single dose level should be the maximum human dose without correcting for body weight (i.e., 1 human dose = 1 animal dose)" (European Medicines Agency, 2020). Accordingly, the 0.3 mL mRNA vaccine and 0.5 mL inactivated vaccine, which are the standard clinical doses for humans, were administered in our animal model without adjustment for body weight.

However, we acknowledge that rodents have significantly smaller body mass and different pharmacokinetics compared to humans, which could influence antigen exposure, immune response intensity, and local tissue reactions. While using the full human dose without adjustment is a standard approach in regulatory vaccine toxicology studies, it may not fully account for species-specific pharmacokinetic and immunogenic differences. Therefore, we reComment that future studies evaluate dose-scaling approaches to improve translational relevance.

A more biologically appropriate dose could be determined by applying the allometric scaling approach proposed by Nair & Jacob (2016). Based on a human-to-rat dose conversion factor of approximately 6.2, the 0.3 mL human dose would corResponse to reviwer to approximately 0.05 mL in a 250 g rat.

To address these limitations, future studies should include pharmacokinetic (PK) analyses to assess antigen bioavailability, immune response intensity, and systemic distribution. Additionally, comparative studies with different dose groups should be conducted to establish a clearer dose-response relationship.

We appreciate your valuable feedback. To enhance the accurate evaluation of vaccines in small animal models, we emphasize the importance of optimizing dose selection. Future research should prioritize species-specific dose adjustments to improve translational validity.

So, We added the explanation of the determination of vaccine dose in the ‘’Materials Methods, Vaccination Proccedure’’

References:

  • European Medicines Agency (EMA). (2020). ICH S5 (R3) Guideline on Detection of Reproductive and Developmental Toxicity for Human Pharmaceuticals.
  • Nair, A. B., & Jacob, S. (2016). A simple practice guide for dose conversion between animals and humans. Journal of Basic and Clinical Pharmacy, 7(2), 27-31.

Comment 3:

Why did the control rats not receive diluent? 

 The appropriate control for the study would be to administer the same volume of the diluent to the control rats rather than providing no injection or an alternative control

Response to reviwer 3:

Thank you for your valuable feedback regarding the control group. The control rats did receive a diluent, 0.9% NaCl solution. We added this explanation in ‘’Materils and Methods, Vaccination Procedure’’.  

 General comments

While the study presents a range of histological findings following the administration of a full human dose of vaccine in rodents, several critical concerns limit the generalisability and relevance of these results to human health. 

 First, administering an unscaled human dose in a rodent model disregards established allometric scaling principles, raising questions about whether the observed histological changes are a function of excessive dosing rather than the intrinsic toxicity of the vaccine itself. This dose could be up to 40 times a human dose.

 Second, given the substantial evidence from both clinical and preclinical studies indicating that mRNA and inactivated vaccines do not negatively affect fertility or reproductive health in humans, the relevance of these histological outcomes to the broader scientific discourse is questionable. The findings may represent artefacts of the dosing strategy rather than a true reflection of vaccine safety. Therefore, while the histological data are detailed, their interpretation in the context of human vaccine safety should be approached with caution, and further studies employing appropriately scaled doses and comprehensive functional assessments are needed to validate these observations.

 Administering a full human dose of an mRNA vaccine to a rat is likely to induce a hyperinflammatory state both locally and systemically, with histological evidence of oedema, immune cell infiltration, and potential tissue damage. These responses underscore the importance of appropriate dose scaling in preclinical models to ensure that the observed outcomes accurately reflect the vaccine’s safety under conditions that are relevant to human use. I suggest modelling this on the preclinical methods and if not then justify to the reader.

Response to reviwer to general comment:

Thank you for your feedback. Our study was conducted in accordance with the ICH S5 (R3) guidelines, which reComment administering the full human dose without body weight correction. However, we acknowledge that this approach does not account for allometric scaling principles and may have induced an exaggerated immune response in rodents. Future studies should apply allometric scaling, as suggested by Nair & Jacob (2016), to achieve a more biologically relevant dose. Additionally, pharmacokinetic analyses, cytokine profiling, and immune response assessments should be incorporated to better interpret the vaccine’s histological effects. While clinical and preclinical evidence indicates that mRNA and inactivated COVID-19 vaccines do not adversely affect human reproductive health, our study aims to explore potential mechanistic pathways rather than make direct conclusions about human risk. Therefore, we reComment that future research include lower-dose groups, long-term hormonal and fertility assessments, and a control group of naturally infected animals. These limitations have been clearly addressed in the discussion, emphasizing the need for cautious interpretation of our findings in the context of human vaccine safety. We appreciate your valuable insights.

Reviewer 3 Report

Comments and Suggestions for Authors

Is a well written manuscript with relevant information for scientific community and general population.

Minor comments

Line 374-375

Is missing text?

Figure 6

y-axis might be said “serum AMH levels” not mean …

Author Response

Comment 1: Line 374-375

Is missing text?

Response to reviwer 1:

Thank you very much for your warning. Typing mistake was corrected.

Comment 2:

Figure 6

y-axis might be said “serum AMH levels” not mean …

Response to reviwer 2:

In Figure 6, the y-axis was changed according to you suggestion.

Round 2

Reviewer 1 Report

Comments and Suggestions for Authors

 The manuscript has been duly corrected according to the suggestions made to the authors.